# Effect of Bismuth Ferrite Nanometer Filler Element Doping on the Surface Insulation Properties of Epoxy Resin Composites

**DOI:** 10.3390/nano11092200

**Published:** 2021-08-26

**Authors:** Jun Xie, Chaoxuan Xiao, Shuai Shao, Qijun Duan, Qing Xie, Fangcheng Lü

**Affiliations:** 1Hebei Provincial Key Laboratory of Power Transmission Equipment Security Defense, North China Electric Power University, Baoding 071003, China; junxie@ncepu.edu.cn (J.X.); XCX@ncepu.edu.cn (C.X.); duan_ncepu@163.com (Q.D.); lfc0818@sohu.com (F.L.); 2State Grid Changchun Power Supply Company, Changchun 130021, China; shhhhhaoshuai@163.com

**Keywords:** bismuth ferrite nanoparticles, epoxy resin, DC flashover voltage, surface charge, trap distribution

## Abstract

In the direct current electric field, the surface of epoxy resin (EP) insulating material is prone to charge accumulation, which leads to electric field distortion and damages the overall insulation of the equipment. Nano-doping is an effective method to improve the surface insulation strength and DC flashover voltage of epoxy resin composites. In this study, pure bismuth ferrite nanoparticles (BFO), as well as BFO nanofillers, which were doped by La element, Cr element as well as co-doped by La + Cr element, were prepared by the sol-gel method. Epoxy composites with various filler concentrations were prepared by blending nano-fillers with epoxy resin. The morphology and crystal structure of the filler were characterized by scanning electron microscopy (SEM) and X-ray diffraction (XRD) tests. The effects of different filler types and filler mass fraction on the surface flashover voltage, charge dissipation rate, and trap characteristics of epoxy resin composites were studied. The results showed that element doping with bismuth ferrite nanofillers could further increase the flash voltage of the composites. The flashover voltage of La + Cr elements co-doped composites with the filler mass fraction of 4 wt% was 45.2% higher than that of pure epoxy resin. Through data comparison, it is found that the surface charge dissipation rate is not the only determinant of the flashover voltage. Appropriately reducing the surface charge dissipation rate of epoxy resin composites can increase the flashover voltage. Finally, combining with the distribution characteristics of the traps on the surface of the materials to explain the mechanism, it is found that the doping of La element and Cr element can increase the energy level depth and density of the deep traps of the composite materials, which can effectively improve the flashover voltage along the surface of the epoxy resin.

## 1. Introduction

Composite insulators based on the epoxy resin (EP) have excellent electrical insulation and thermodynamic properties and are widely used in high-voltage power transmission equipment, such as gas-insulated switchgear (GIS) and large-scale pulse power devices [1,2,3]. Compared with AC GIS, with the continuous improvement of voltage level, the flashover voltage on the insulator surface under DC working conditions is significantly reduced [4,5]. The further improvement of the insulation level of EP composite material has become an important factor to ensure the safe and stable operation of electrical equipment [6].

Studies have found that the surface flashover discharge of epoxy resin composite insulators is a key factor leading to insulation failures of electrical equipment [7], but the mechanism of surface flashover discharge has not been fully revealed. A large number of experimental data show that the gas–solid interface of insulating materials will accumulate a large number of charges under the action of DC electric field or partial discharge at the electrode and gas–solid interface [8,9]. These charges easily overflow the material surface under the action of the electric field, inducing surface flashover [10,11], which reduces the insulation reliability of equipment and even leads to insulation failure. For this reason, the researchers managed to improve the surface charge transport capacity of epoxy resin composites, accelerate the dissipation rate of surface charges, and then increase the surface flashover voltage [12]. However, the high surface conductivity will cause discharge on the surface of the material, which will seriously reduce the insulation characteristics and service life of epoxy resin composite insulation materials. Therefore, although the correlation between the material surface charge dissipation rate and flashover voltage has been proposed in the current studies, the theory is not yet complete, and the experimental studies on the flashover characteristics of materials with different conductivity also need to be further verified. In addition, Li, S.T. et al. proposed that the trap energy level and trap distribution on the surface of polymer materials are also important factors affecting flashover [13]. The physical defects in the process of material synthesis and use will produce shallow traps, which are helpful for carrier transport, while the deep traps caused by chemical defects trap the charge, inhibiting the carrier movement. Yu, K.K. et al. found that the existence of shallow traps on the surface makes the surface charge of the dielectric easily excited to escape from the trap, resulting in a decrease in flashover voltage along the surface [14]. However, Li, S.T. and Shao, T. et al. pointed out that if the depth of the dielectric trap is too deep, the charge accumulated in the trap will cause serious distortion of local field strength and store a large amount of polarization energy. When subjected to external interference, the flashover voltage will be reduced [13,15]. Therefore, the explanation of flashover by trap theory still needs to be further improved, and the method of increasing the flashover voltage by adjusting the distribution of material traps needs in-depth study.

Lewis, T. J. proposed the concept of nano-dielectric in 1994 [16]. The doping modification of inorganic nanoparticles can change the trap energy level and conductivity of epoxy resin composites and effectively improve the charge dissipation ability and flashover voltage of epoxy resin composites [17,18]. He, J.L. et al. studied the interface effect between the filler and matrix, and the interaction between the nanoparticles and the polymer molecular chains made the interface have a particular structure and properties, which affected many electrical properties of the material [19].

Semiconductor nano-materials have a narrower bandgap than insulators. The preparation of epoxy resin composites by doping them can reduce the energy barrier overcome by carrier migration in traps, increase the rate of charge dissipation on the surface of the material, and reduce the internal electric field distortion [20]. At present, there are relatively few studies on the effect of semiconductor nanofillers with different morphologies on the surface insulation properties of epoxy resin composite materials. One-dimensional nanofibers have a high specific surface area and aspect ratio, as well as special-shaped atoms, electronic structure and other properties, which will affect the comprehensive properties of composites [21]. Chi, Q. studied the effect of nano-TiO_2 _morphology on the breakdown field strength and space charge characteristics of polyethylene composites. The results show that nano-TiO_2 _ fiber doped composites have higher breakdown field strength than nano-TiO_2 _ particles. In addition, the finite element simulation results show that the space charge accumulation and electric field distortion of the nanofiber doped composite materials are significantly reduced [22]. Zhang, D. used multifunctional carbon nanofibers (CNF) doped with epoxy resin. The modified composites have good sand corrosion resistance, and the flexural strength of the composites is increased by 23.8% by carbon fiber [23]. Bismuth ferrite (BiFeO_3_, BFO), a semiconductor material, is widely used in digital storage devices, magnetoelectric induction equipment and other fields [24]. BFO materials theoretically have unexpectedly large remanent polarization [25]. In the experimental process of preparing BiFeO_3,_ the Bi element is easy to volatilize, and Fe^3+^ will be transformed into Fe^2+^. These factors cause the leakage current density of the material to increase, making it difficult to polarize the material. Therefore, the application of BFO materials is limited [26]. Yan F et al. studied the effect of A-site doping of lanthanide elements La^3+^ on the performance of BiFeO_3_ films and prepared pure BiFeO_3_. The density of the doped film was increased, and the leakage current was decreased [27]. Zhang Y et al. studied the effect of B-site doping of Cr^3+^ on the properties of BiFeO_3_ materials. It was found that the dielectric constant of BiFeO_3_ was increased, the dielectric loss was decreased, and the leakage current density was decreased. However, as the proportion of element doping increases, the properties of BiFeO_3_ materials decrease immediately [28]. Currently, there are relatively studies on epoxy resin modified by semiconductor nano-filler doped with bismuth ferrite nano-filler elements. The effect of doping of bismuth ferrite nano-filler elements on the surface insulation properties of epoxy resin composites is worthy of in-depth study.

In this study, pure BFO nanofillers and BFO nanofillers doped with La and Cr were prepared by the sol-gel method, and EP composite samples with different filler mass fractions were prepared. The modification effect was characterized by scanning electron microscopy (SEM) and X-ray diffraction (XRD) tests. Negative polarity DC flashover and charge dissipation rate were measured. The effect of doping of bismuth ferrite nano-filler elements on the surface insulation properties of epoxy resin composites was analyzed. Based on the trap distribution characteristics, the variation law of surface insulation properties of materials is analyzed. The results can provide a certain reference for further improving the surface insulation performance of the EP composite.

## 2. Materials and Methods

### 2.1. Materials

The main raw materials for this experiment are:

Bisphenol A diglycidyl ether (DGEBA, E51), epoxy value 0.5~0.54 mol/100 g, methyl-tetrahydro phthalic anhydride (MTHPA,504), the mass fraction of anhydride is ≥41%; 2,4,6-tris (dimethyl aminomethyl) phenol (DMP-30), the amine value is 590~650 mg/g, produced by Shanghai Resin Factory; bismuth nitrate pentahydrate (Bi(NO_3_)_3_·5H_2_O), iron nitrate nonahydrate (Fe(NO_3_)_3_·9H_2_O), lanthanum nitrate hexahydrate (La(NO_3_)_3_·6H_2_O), chromium nitrate nonahydrate (Cr(NO_3_)_3_·9H_2_O), Tartaric acid (TA, C_4_H_6_O_6_). The above reagents are all analytical reagents (AR) and purchased from Aladdin Chemical Reagent Co., Shanghai, China.

### 2.2. Preparation of BiFeO_3_ Nanoparticles

In this paper, the pure bismuth ferrite nanoparticles (BiFeO_3_, BFO) nanofiller was prepared by the sol-gel method, and the specific steps are as follows:Take Bi(NO_3_)_3_·5H_2_O and Fe(NO_3_)_3_·9H_2_O with a molar ratio of 1:1 and dissolve them in ethylene glycol with a concentration of 1mol/L;Add tartaric acid (TA) to the solution, and the molar ratio of Bi(NO_3_)_3_·5H_2_O, Fe(NO_3_)_3_·9H_2_O and TA is 1:1:1, and the magnetic agitator was used to stir the mixture for 12 h at room temperature to make the nitrate fully dissolved. Third bullet;Transfer the mixture to the fume hood and stir in the oil bath at 60 °C for 36 h. After full volatilization of glycol, the solvent can form a dry gel;Grind the dry gel into powder, place it in a quartz boat, and push it into a tubular furnace for heating at a heating rate of 5 °C/min. After holding at 120 °C for 1 h, 300 °C for 2 h, and 600 °C for 2 h, the dry gel is extracted and quickly cooled to room temperature;Wash twice with dilute nitric acid with 10% volume fraction (to remove the Bi oxide or Bi salt overproduced by Bi(NO_3_)_3_), and then wash twice with deionized water (to remove the nitrate ion introduced by nitric acid).

The pure BFO nanofillers can be obtained by the above process.

The preparation method of elemental doped BFO nanofillers was the same as that of pure BFO nanofillers, except for some drug dosage differences: 5% La element doping is A-site doping, and part of Bi(NO_3_)_3_·5H_2_O was replaced by La(NO_3_)_3_·6H_2_O, and the molar ratio of the two was 0.05:0.95. In preparation, 3% Cr element doping is B-site doping, Cr(NO_3_)_3_·9H_2_O was used to replace part of Fe(NO_3_)_3_·9H_2_O, and the molar ratio of the two was 0.03:0.97. A total of 5% La + 3% Cr element doping is A-site and B-site co-doping, La(NO_3_)_3_·6H_2_O, Bi(NO_3_)_3_·5H_2_O, Cr(NO_3_)_3_·9H_2_O and Fe(NO_3_)_3_·9H_2_O were added at the same time, and the molar ratio was 0.05:0.95:0.03:0.97.

To compensate for the Bi element volatilized during the heating process, the Bi(NO_3_)_3_·5H_2_O was added in excess by 10% during the preparation of nanoparticles. In addition, for the convenience of description, the BFO nanofillers doped with 5% La, 3% Cr and 5% La + 3% Cr were denoted as BL5FO, BFC3O, and BL5FC3O, respectively.

### 2.3. Preparation of Epoxy Resin Composites

The preparation method of the EP composite sample doped with BFO nano-filler is as follows:The prepared BFO nanofillers were placed in a beaker, and DGEBA was added. The BFO nanofillers were mechanically stirred for 60 min in an oil bath environment at 60 °C and dispersed by ultrasonic for 30 min to make the fillers fully dispersed in the resin;The curing agent MTPHA and accelerator DMP-30 were added, and the mass ratio of DGEBA, MTPHA, and DMP-30 was 100:80:1, and the mixed suspension was fully stirred for 30 min and degassed;The mixed suspension was placed in the polytetrafluoroethylene (PTFE) mold, and gradient curing was performed at 80 °C/1 h + 100 °C/10 h. After cooling to room temperature, the mold was stripped to obtain the BFO/EP epoxy resin composite material sample.

According to the above methods, BL5FO/EP, BFC3O/EP, BL5FC3O/EP, and BFO/EP composite samples were prepared with the mass fraction of 1%, 2%, 4%, 8%, and 16%, respectively.

### 2.4. Flashover Voltage and Surface Charge Dissipation Test

The flashover voltage of the prepared EP composite was tested. The flashover voltage test platform is shown in Figure 1.

During the flashover test, the samples are placed in a cylindrical sealed experimental cavity. The outer diameter of the cavity is 300 mm, the height is 350 mm, and the thickness is 8 mm. The inside of the vessel is an air environment. A discharge electrode is installed inside the vessel, and a high-voltage DC power supply is connected externally through the high-voltage bushing. To facilitate the experiment, a two-dimensional mobile device is installed inside the cavity to realize the smooth movement of the sample table. The cavity is equipped with an observation hole, which is convenient for observing the internal discharge phenomenon. The discharge electrode adopts the needle-needle electrode form to simulate the extremely uneven electric field and make flashover easier. The total length of the electrode is 15 mm, the bottom diameter is 4 mm, and the tip inclination angle is 15°. The distance between the high voltage electrode and the ground electrode is 7 mm. The lower surface of the electrode is planar, and the electrode surface is smooth and free of burrs. In this way, the electrode and the sample surface can be completely attached so that the flashover occurs completely on the composite sample surface. In addition to the above equipment, the test platform also includes a high voltage DC power supply (0~-50 kV adjustable), high voltage measurement probe (Tektronix, P6015A, 1000:1, San Francisco, CA, USA), digital oscilloscope (Tektronix DPO 2002B, San Francisco, CA, USA), protection resistance (1 MΩ), etc. The voltage variation can be obtained by a high voltage measuring probe and digital oscilloscope. The protection resistor can prevent excessive current during flashover, which may cause harm to persons and equipment.

The specific steps of the flashover test on EP composite samples are as follows:Put the EP composite material in deionized water and clean it with an ultrasonic cleaner for 10 min;Put the sample in the drying oven, and wear disposable gloves to transfer the sample after the sample is completely dry;The sample is placed on the sample table, and operate the two-dimensional moving device by placing the needle-needle electrode in the center of the sample with good contact;Start the high-voltage power supply with a pressure gradient of 100 V/s until flashover occurs along the surface.

The flashover that occurs is judged by observing the waveform of the oscilloscope and the arc between the electrodes. Adjust the voltage to zero immediately after flashover, and conduct the next surface flashover test after an interval of 60 s. Each group of samples is measured 10 times, and the average value is recorded as the surface flashover voltage of the sample.

The schematic diagram of the test platform for charge dissipation characteristics is shown in Figure 2. The measuring system uses a high-voltage DC power supply to make the charging pin generate corona discharge above the sample and inject charge into the surface of the test sample. The active attenuation probe (Trek 3455ET, Bavaria, Germany) was used to detect the surface point potential information. The potential was measured by an electrostatic potentiometer (Trek P0865, 10 kV, Bavaria, Germany), and the measured results were stored in a data acquisition card (Altai USB3202, Beijing, China). The specific steps for the charge dissipation characteristics test are as follows:Place the sample to be tested in deionized water, ultrasonically clean it for 10 min, and then dry it in a drying oven at 60 °C;Place the dried sample on the stage and perform a surface charge test on the surface of the sample. If the surface potential of the sample is not zero, the sample can be continued to be placed in the 60 °C drying oven until the surface potential of the sample is 0. In this way, it is ensured that no charge is accumulated on the sample surface;Move the charging pin above the surface of the sample, and the charging pin is 2 mm away from the sample. Apply a negative DC voltage with an amplitude of 7 kV to the charging pin, and charge the sample for 1 min;Remove the charging pin immediately after charging is completed.

Through a two-dimensional operation device, the probe was transferred directly above the charging position, and the acquisition card was opened to collect potential data every 100 ms. The average value of 10 potential data was calculated and output [29]. The experimental time of charge dissipation is 30 min. After the test is completed, the exponential decay curve of the potential data corresponding to the time can be obtained.

## 3. Results and Discussions

### 3.1. Characterization of BFO Nano-Filler and EP Composite

Figure 3 shows the SEM test [30] results of pure phase BFO nanoparticles and BL5FO, BFC3O, and BL5FC3O nanoparticles. According to the results in Figure 3, the particle size distribution of pure BFO nanoparticles is uneven, and the particle surface has obvious prismatic morphology, which may be the result of the preferred growth of crystal planes [31].

The element doping makes the filler particles more uniform, and the particle size is further reduced. The decrease in particle size may be caused by the substitution of Cr^3+^ and La^3+^ (with smaller ion radius) for Fe^3+^ and Bi^3+^ (with larger ion radius). When the elements with a smaller ionic radius are incorporated into the lattice, the lattice will be deformed and contracted, resulting in a denser network [32,33,34]. The study found that the smaller the particle size of the filler, the lower the flashover voltage of BFO/EP composites. However, considering the overall performance changes of the composite material, the BFO filler particle size has little effect on the epoxy resin composite material [35].

As shown in Figure 4, to study the lattice structure of the experimentally prepared BFO nanofillers, X-ray diffraction (XRD) tests [36] were performed on each nanofiller. $C_{u}K^{\alpha }\left(\lambda =1.5418Å\right)$ was used as the ray source with the scanning range of 20°~80°, and the scanning interval was 0.02°.

From the results in Figure 4, it can be seen that the XRD diffractograms of each nanofiller are consistent with the standard diffractogram of BiFeO_3_. The characteristic diffraction peaks of crystal planes such as (012), (104), (110), (006), (024), (116), (018) and (220) appear in the X-ray diffraction patterns, which can confirm that the sample prepared is rhombohedral perovskite structure with space group R3c [37], and all the diffraction peaks are relatively sharp, indicating that the obtained samples have good crystallinity [38]. There is no impurity peak in BL5FO, but there is a small amount of impurity phase in the two samples of BFC3O and BL5FC3O, as shown by “*”, which is mainly impurity phase Bi_2_Fe_4_O_9_ [39]. However, some impurity phases are free of the unit cell, and diffraction peaks are not regularly reflected in the XRD diffractogram (as shown by “♦” in the figure). These diffraction peaks are difficult to match with the material library, so their properties cannot be identified. It is worth noting that most phases in the compounds of BL5FO, BFC3O and BL5FC3O have the same structure as BFO, i.e., R3c, which means that the substitution of La and Cr elements does not affect the crystal structure of the parent BFO compound, which is significant if the ferroelectric properties of pure BFO are to be maintained [40].

Figure 5 is the energy dispersive spectrometer (EDS) [41] spectrum of BL5FO, BFC3O, and BL5FC3O, which is used to detect the element composition and content analysis of the material composition. The energy spectrum results show that La and Cr elements have been successfully doped into the lattice structure of BFO.

EP composites doped with BFO nanoparticles were prepared according to the method in Section 2.3. Figure 6 shows the SEM test results of EP composites doped with pure phase BFO nano-filler. As shown in Figure 6, with the increase in the mass fraction of the filler, the density of the filler in the matrix increases and agglomeration occurs. When the mass fraction of filler is 2% and 8%, the nano-filler in EP composite is more uniform. As shown in the red circle in Figure 6c, when the mass fraction of filler was 16%, the nanoparticles in EP composite showed obvious agglomeration.

### 3.2. Experimental Results of Flashover Voltage

Figure 7 shows the flashover voltage test results. The ambient temperature of the flashover voltage test is 27 °C, and the relative humidity is 30%. According to the results in Figure 7a–d, the flashover voltage of the EP composite after doping the nano-filler is more complex: with the increase in the mass fraction of the filler, the flashover voltage of the BFO/EP composite shows an “M” shaped change rule. The flashover voltage of BL5FO/EP composites changes in an inverted “V” shape. The flashover voltage of BFC3O/EP and BL5FC3O/EP composites shows an “N” shaped change rule. The yellow dotted line in Figure 7e is the flashover voltage of the pure EP system. According to the figure, the flashover voltage of each EP composite material is higher than that of a pure EP system when the mass fraction of the nano-filler is less than 16%.

The analysis shows that the introduction of nano-fillers can provide a path for the surface charge to dissipate, thereby reducing the degree of electric field distortion on the surface of the composite and increasing the value of the flashover voltage. The influence of nano-filler doping concentration on EP composites is more complicated: with the increase in doping concentration, the distance between nano-filler decreases, the number of charge dissipation channels increases, and the flashover voltage increases. However, increasing the doping concentration will reduce the uniformity of the dispersion of nanoparticles in the matrix, resulting in regional differences in parameters, such as surface conductance and traps on the EP composite surface. Increasing the doping concentration will also increase the distortion of the surface electric field and reduce the flashover voltage. In addition, as the doping concentration increases, the interface between the nanoparticles and the polymer begins to overlap, and the scattering of carriers is enhanced. Meanwhile, the agglomeration of nanoparticles causes new defects in the materials [38]. In turn, the surface flashover voltage of the EP composite decreases. Under the combined effect of the above-mentioned reasons, the flashover voltage of EP composites has a complicated change pattern, as shown in Figure 7.

Comparing the results in Figure 7, the flashover voltage of EP composite filled with element doped BFO nanofiller is higher than that of pure BFO nano-filler. At 4 wt%, the flashover voltage of BL5FC3O/EP composite has been increased by 45.2% compared with the pure EP system, which has a good application prospect.

### 3.3. Effect of BFO Element Doping on Charge Dissipation Characteristics of EP Composites

Under DC conditions, a large amount of charge accumulates on the surface of the insulator, which causes electric field distortion at the gas –solid interface, leading to surface flashover failure [8]. Therefore, it is necessary to study the surface charge dissipation characteristics of EP composites to analyze the flashover process of composites.

Figure 8 shows the charge dissipation curves of BFO/EP and BL5FO/EP composites with different filling mass fractions. According to the results in Figure 8, the addition of a nano-filler can improve the charge dissipation rate of the composites. At a lower filling concentration, the charge dissipation rate of the composite is significantly higher than that of the pure EP system. The conductivity of the composite material increases with the increase in nano-filler concentration, and the dissipation rate of surface charge should be increased. However, the high concentration of nano-filler causes the agglomeration of filler in the matrix, which leads to new defects. Therefore, when the packing concentration is 16 wt%, the charge dissipation rate of the BFO/EP composite decreases. In addition, according to the results of Figure 7 and Figure 8, for BFO/EP composites, the charge dissipation rate of the composite with 16 wt% filling concentration is higher than that of 2 wt% and 4 wt% filling concentration, but the flashover voltage is only 10.6 kV. For BL5FO/EP composites, the charge dissipation rate increases with the increase in filling concentration, but the flashover voltage changes in an inverted “V” shape, which increases first and then decreases. Therefore, it is not that the greater the surface charge dissipation rate of the composite, the higher its flashover voltage. Appropriately reducing the dissipation rate of surface charge will help increase the flashover voltage.

Figure 9 shows the surface charge dissipation curves of EP composites doped with various nano-fillers at the filling concentration of 4 wt%. It can be seen from the figure that the surface charge dissipation rate of EP composites can be reduced by doping BFO nanoparticles with elements.

The analysis shows that the pores and cracks of the material itself make the surface of the material have voids; the reduction of Fe^3+^ to Fe^2+^ and the volatilization of Bi elements will cause oxygen vacancies in the material. The above factors make the conductivity of the pure phase BFO sample higher, and the surface charge dissipation rate of EP composites is higher under the same doping concentration [26]. The doping of La and Cr helps to reduce the conductivity of BFO nanoparticles, which is mainly due to the following reasons: Firstly, the doping of high valence Cr in the material controls the valence fluctuation of Fe and reduces the oxygen vacancy. Secondly, the La–O bond is more difficult to break than the Bi–O bond (the dissociation energy of the La–O bond is 799 ± 13 kJ/mol; the dissociation energy of the Bi–O bond: 343 ± 6 kJ/mol) [42]. The volatilization of Bi can be inhibited, and the perovskite structure can be stabilized by replacing part of Bi with La element. Finally, La and Cr doping can improve the surface morphology of BFO, increase the density of the material network, and the reduction of voids is helpful to reduce the leakage current density. Therefore, the dissipation rate of surface charge of EP composite decreases after La doping, Cr doping, or La + Cr co-doping.

### 3.4. Effect of BFO Element Doping on the Trap Characteristics of EP Composites

The trap structure can affect the charge transport in the material system by consolidating and constraining the charge. If the charge breaks free of the trap, it can cause a large amount of energy release and induce flashover. Therefore, by studying the trap characteristics of composites, the mechanism of flashover voltage increase in EP composites can be explored [43,44].

At present, the characterization methods of material trap characteristics mainly include thermally stimulated current (TSC) [45], isothermal surface potential decay (ISPD) [38], pulsed electro-acoustic (PEA) [46], and laser-induced pressure pulse (LIPP) [47], etc. The ISPD method is widely used because of its simple experimental platform and convenient operation.

Based on the basic theory of the ISPD method [48], the trap energy level can be expressed as:(1)$$E_{T}=k_{B}T\ln\left[t\frac{{\left(k_{B}T\right)}^{3}}{6h^{3}v^{2}}\right]$$
(2)$$Q_{s}(t)=t\frac{4\varepsilon_{0}\varepsilon_{r}}{eL^{2}k_{B}T}\left|\frac{dV_{s}(t)}{dt}\right|$$
where, *E_T_* is the trap level; *k_B_* is the Boltzmann constant; *T* is the absolute temperature of the test environment, in K; *h* is Planck constant; *ν* is the natural vibration frequency around the defect point at the orthogonal plane of the moving direction; *Q_S_*(*t*) is the trap charge density; $\varepsilon_{0}$ is the vacuum dielectric constant; $\varepsilon_{r}$ is the relative dielectric constant; *d* is the thickness of the sample; *V_s_*(*t*) is the surface potential of the sample.

Studies have shown that the surface potential attenuation curve of the sample can generally be fitted based on the double exponential function, and its expression is:(3)$$V_{s}(t)=A_{1}\cdot e^{-t/\tau_{1}}+A_{2}\cdot e^{-t/\tau_{2}}$$
where *A*_1_, $\tau_{1}$ and *A*_2_, $\tau_{2}$ are fitting parameters. After fitting analysis, the characteristics of trap energy level distribution can be obtained based on Equations (1) and (2).

Figure 10 shows the trap energy level distribution characteristics of the pure EP system and the EP composite filled with various nano-fillers (filling concentration is 4 wt%). The two peaks on each curve represent the shallow trap (Area A) and the deep trap (Area B), respectively.

Figure 10 shows that the shallow trap energy level depth, shallow trap energy level density, and deep trap energy level density of BFO/EP composite are close to those of the pure EP system. However, the deep trap energy level depth of BFO/EP composites is significantly higher than that of pure EP composites. Compared with BFO/EP composites, there is no significant change in the shallow trap energy level depth of BL5FO/EP, BFC3O/EP, and BL5FC3O/EP composites doped by elements. However, the shallow trap energy level density decreases, while the deep trap energy level depth and deep trap energy density both increased. The BL5FC3O/EP composites Co-doped with La and Cr only contain deep trap peaks, and the depth and density of deep trap energy levels are the highest.

When applied voltage, the local field strength at the junction of electrode, gas, and sample surface is high, and the primary electron will be first ionized and formed there. Some primary electrons are easily absorbed by the polymer surface and form a space charge, which distorts the electric field. The other part of the primary electrons will cause the emission of secondary electrons under the action of the distorted electric field, which will lead to electron collapse and eventually flashover [49,50,51].

The shallow trap has a lower energy level, and the bound charge is easy to collapse. The level of the deep trap is deep, and the bound charge is difficult to escape. Therefore, shallow traps are easier to improve the charge dissipation rate than the deeper traps and improve the uniformity of the electric field. Compared with BFO/EP composites, the shallow trap level depth of the EP composites does not change significantly after the BFO is doped with La, Cr, or co-doped with La and Cr elements. However, the depth and density of the deep trap energy level of the element-doped composites are both increased, and the depth and density of the deep trap energy level of the BL5FC3O/EP composite are the highest, which is also consistent with the test results of charge dissipation characteristics in Figure 9.

However, according to the above analysis results, although charge dissipation characteristics affect the flashover voltage, they are not the key factors. As can be seen from Figure 10, each sample has obvious characteristics of deep traps. Further analysis shows that after the electrons in the shallow trap collapse and escape, it is easy to trigger the emission of secondary electrons and promote the occurrence of flashover. Deep traps make it difficult for electrons to escape. Increasing the energy level and density of deep traps can reduce the number of primary electrons, make it difficult to form secondary electron emissions on the material surface, and thus increase the flashover voltage [52]. Compared with BFO/EP composites, the depth and density of deep trap energy level and flashover voltage of EP composites are increased after the BFO is doped with La, Cr, or co-doped with La and Cr elements.

## 4. Conclusions

In this paper, bismuth ferrite (BFO) nanofillers doped by La, Cr, and La + Cr elements were prepared and filled into epoxy resin to prepare epoxy resin (EP) composites. In addition, the insulation characteristics of the EP composite surface were tested and analyzed. The main conclusions are as follows:

The flashover voltage of the EP composite can be further improved by doping BFO nano-filler with La and Cr elements. With the concentration of 4 wt% La + Cr co-doped BFO nano-filler, the flashover voltage of EP composite can reach 14.52 kV. Compared with pure EP material, the flashover voltage of this composite material is increased by 45.2%, which has a better application prospect.

Compared with the BFO nanofillers without element doping, the surface charge dissipation rate of EP composites is decreased after adding La and Cr doped BFO nanofillers. Therefore, the surface charge dissipation rate is not the only determinant of the flashover voltage. Appropriately reducing the surface charge dissipation rate of epoxy resin composites can increase the flashover voltage.

After the BFO nanofiller is doped with La and Cr elements, the deep trap energy level depth and energy level density of the EP composite material are further increased, which is helpful to suppress the secondary electron emission process and increase the flashover voltage. When the filling concentration is 4 wt%, the EP composite filled with La + Cr co-doped BFO nano-filler has the highest deep trap energy level depth and density, as well as the maximum flashover voltage amplitude.

## Figures and Tables

**Figure 1 nanomaterials-11-02200-f001:**
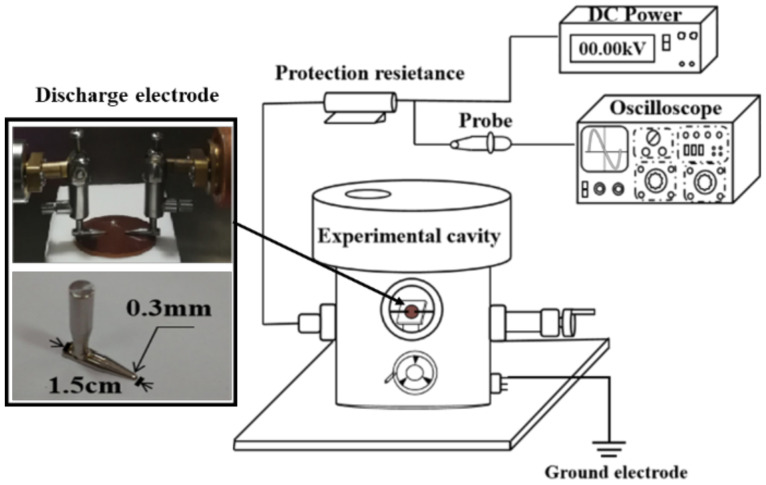
DC flashover experimental platform in the atmospheric environment.

**Figure 2 nanomaterials-11-02200-f002:**
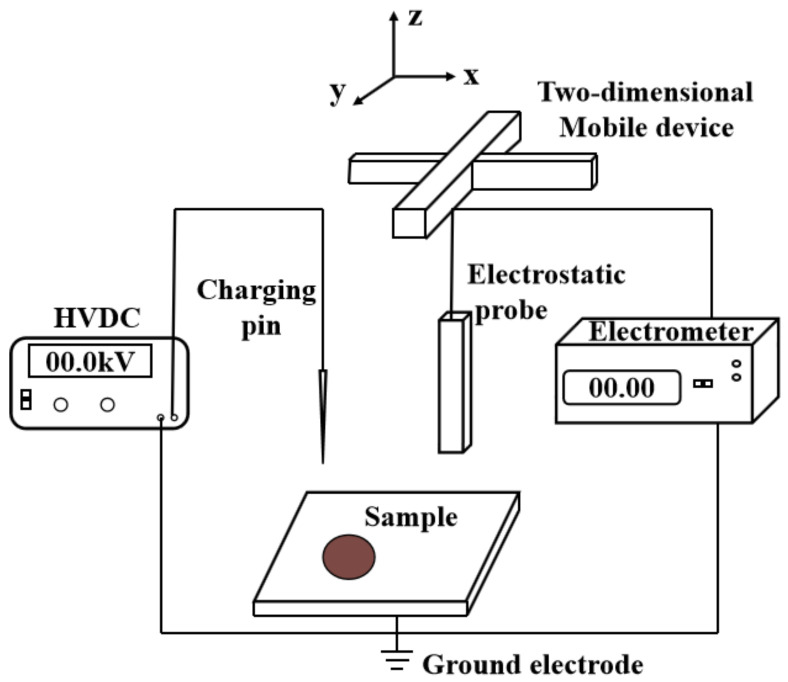
Experimental platform for charge dissipation.

**Figure 3 nanomaterials-11-02200-f003:**
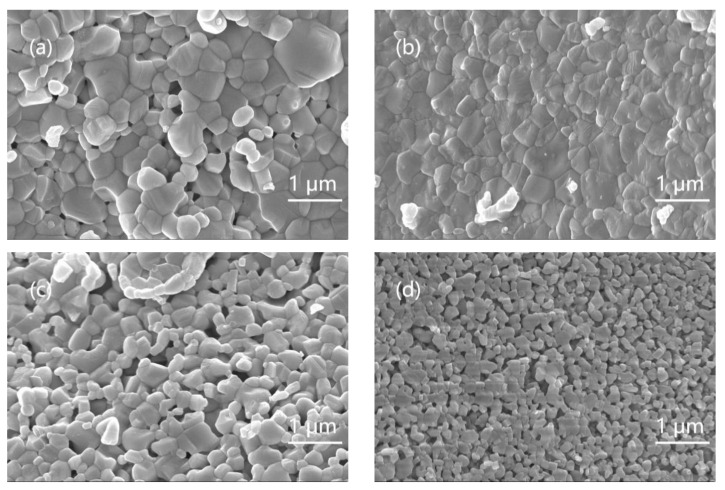
SEM test results of nanofillers and element-doped fillers. (**a**) Pure BFO nanofillers; (**b**) BL5FO nanofillers; (**c**) BFC3O nanofillers; (**d**) BL5FC3O nanofillers.

**Figure 4 nanomaterials-11-02200-f004:**
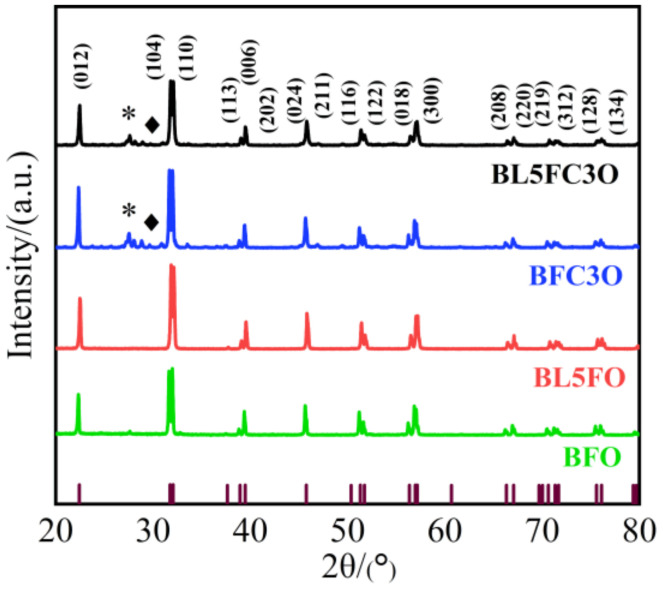
XRD diffractograms of pure BFO and elemental doped BFO. “*” indicates the impurity phase, the main component is Bi_2_Fe_4_O_9_; “♦” indicates the impurity phase free of the unit cell.

**Figure 5 nanomaterials-11-02200-f005:**
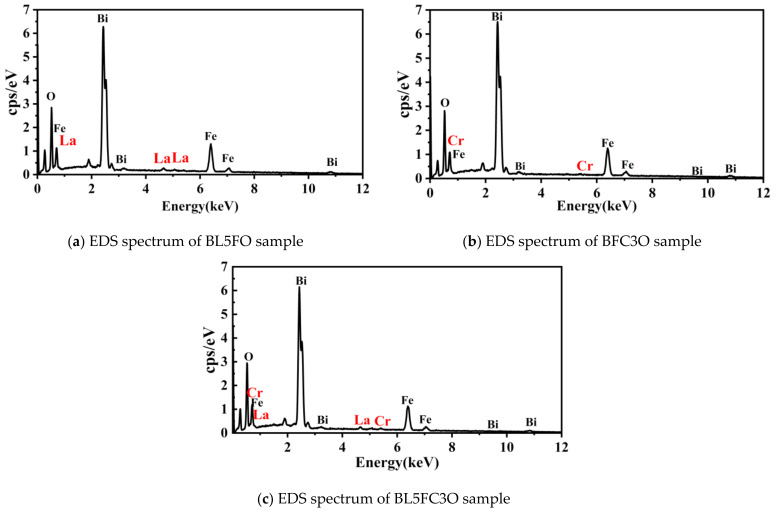
EDS spectrum of BFO sample doped with La and Cr. (**a**) EDS spectrum of BL5FO sample; (**b**) EDS spectrum of BFC3O sample; (**c**) EDS spectrum of BL5FC3O sample.

**Figure 6 nanomaterials-11-02200-f006:**
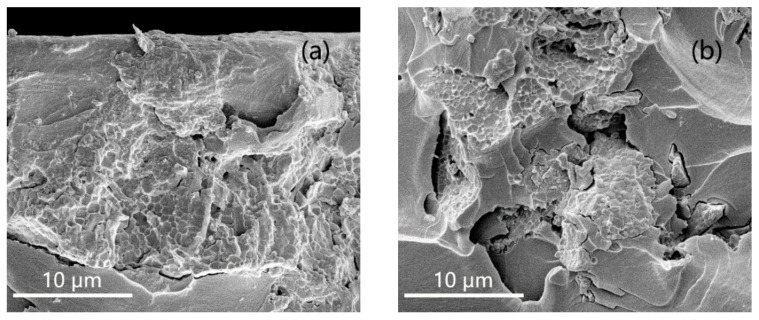
Dispersion of nano-filler in BFO/EP composites. (**a**) 2 wt% BFO/EP; (**b**) 8 wt% BFO/EP; (**c**) 16 wt% BFO/EP.

**Figure 7 nanomaterials-11-02200-f007:**
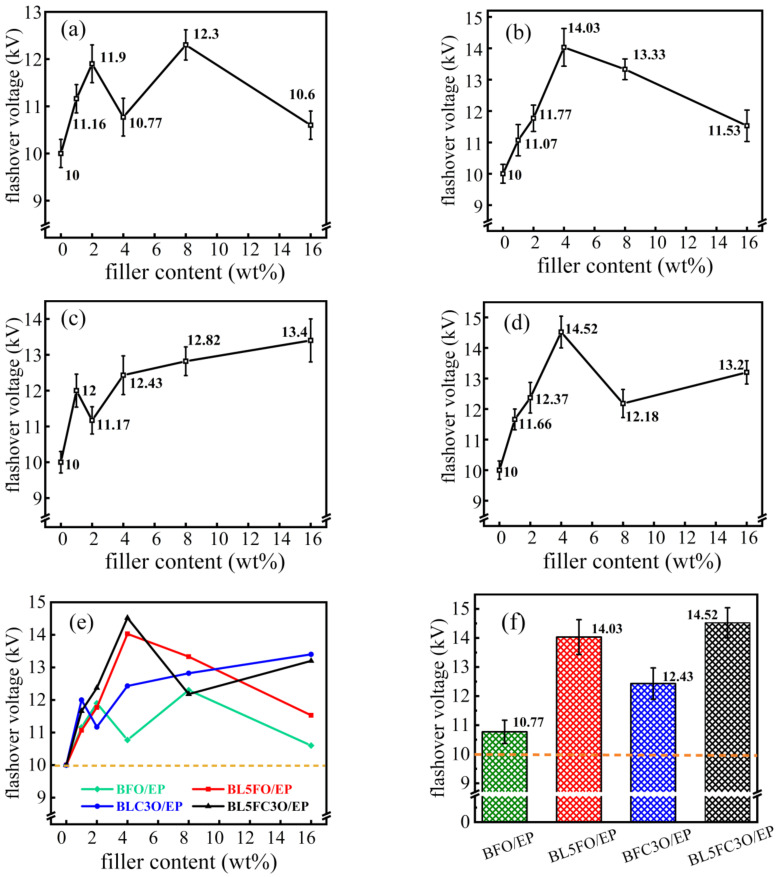
Flashover voltage diagram of element-doped BFO/EP composites with different filler mass fractions. (**a**) BFO/EP; (**b**) BL5FO/EP; (**c**) BFC3O/EP; (**d**) BL5FC3O/EP; (**e**) summary chart; (**f**) the flashover voltage when the composite filler is doped with a mass fraction of 4 wt%.

**Figure 8 nanomaterials-11-02200-f008:**
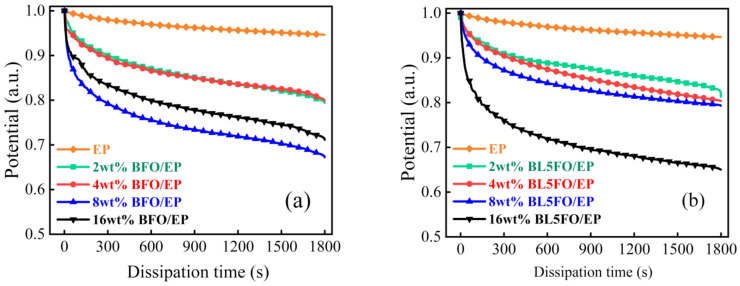
(**a**) Charge dissipation curves of BFO/EP composites with different filling mass fractions; (**b**) charge dissipation curves of BL5FO/EP composites with different filling mass fractions.

**Figure 9 nanomaterials-11-02200-f009:**
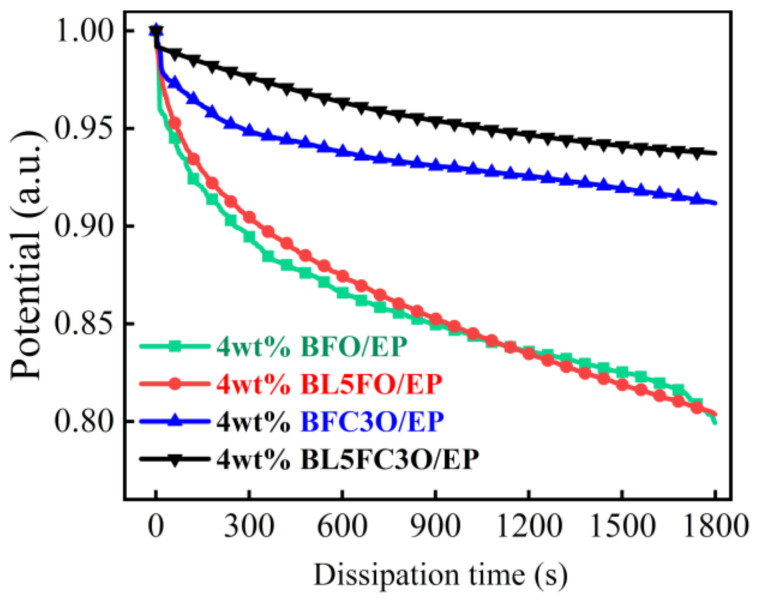
Charge dissipation curve of the composite with doped filler mass fraction of 4wt%.

**Figure 10 nanomaterials-11-02200-f010:**
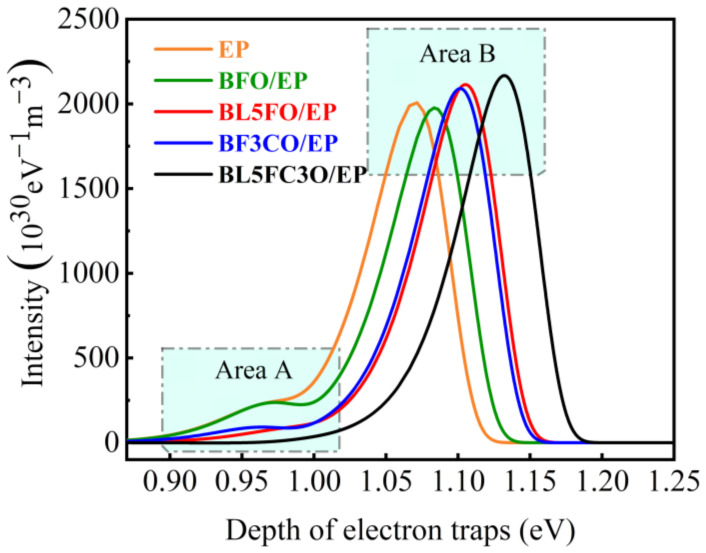
The characteristic curve of trap energy level distribution of EP composite with nano-filler doping (filling concentration of 4 wt%).

## Data Availability

Data are contained within the article.

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
