# Peer review of "Effect of Bismuth Ferrite Nanometer Filler Element Doping on the Surface Insulation Properties of Epoxy Resin Composites"

_nanomaterials, 2021, doi:10.3390/nano11092200_

Round 1

Reviewer 1 Report

The paper is devoted for epoxy resin composites with doped bismuth ferrite nanoparticles preparation and investigations. 
The topic is generally interesting, however the paper contains unexplained places (below) and need major revisions.
Why fillers concentrations were selected only up 16 wt%? For samples BLC3O/EP the flashover voltage tends to increase
at higher fillers concentrations. Maybe this parameter can be improved at higher fillers concentrations?
The dispersion of doped BFO is the same (or similar) as undoped (Fig. 6)? Why SEM pictures are only for composites with undoped BFO?
The size of particles is different for BFO doped with different chemical elements (Fig. 3). What is the impact of particle size on the 
flashover voltage and other composite properties? Can You add some corresponding references?
What is the impact of composites dielectric constant on their insulating properties? 
Abbreviations should by explained by first using and used in all paper text (please check use of EP abbreviation). 
Page 2, line 84, "BFO materials theoretically have higher remanent polarization." Higher, in comparison with which material? 
The reference is needed.
Page 2, line 95 "Currently, there are relatively few studies on the modification of EP with semiconductor nano-filler." What about
investigation of epoxy resin composites with carbon nanotubes, that are widely presented in literature?
In introduction it would be useful to review results about percolative composites insulating properties already published in literature.

Reviewer 2 Report

The Paper is well written and discussed.

Several comments for the paper: Effect of bismuth ferrite nanometre filler element doping on the surface insulation properties of epoxy resin composites

1) Lines  149-153 should be deleted.

2) Fig3 shows no nanofillers. Why do the authors state that BFO is nanoparticles? What is the size of the nanoparticles? Size distribution?

3) SEM, XRD, EDS analysis, and sample preparation details are missing, they should be provided in the Methods section.

4) discussion of the XRD diffractions of BFO is very vague. Please enhance it. Main BFO crystalline peaks should be testified.

Round 2

Reviewer 1 Report

Authors make proper corrections according

to referee remarks. And I suggest to publish the paper as it is.